# Brain health measurement: a scoping review

Angeline Lee ,[1] Suraj Shah,[2] Karyna Atha,[3] Peter Indoe,[4] Naira Mahmoud,[4] Guy Niblett,[4] Vidushi Pradhan,[4] Nia Roberts,[5] Reem Saleem Malouf ,[6] Anya Topiwala [1]

¹Nuffield Department of Population Health, University of Oxford, Oxford, UK
²Health Education England, London, UK
³University of Oxford, Oxford, UK
⁴Health Education Thames Valley (HETV), Oxford, UK
⁵Bodleian Health Care Libraries, University of Oxford, Oxford, UK
⁶Nuffield Department of Population Health, National Perinatal Epidemiology Unit, Oxford, UK

**Correspondence to**
Dr Angeline Lee;
angeline.lee@ndph.ox.ac.uk

## ABSTRACT

**Objectives** Preservation of brain health is an urgent priority for the world's ageing population. The evidence base for brain health optimisation strategies is rapidly expanding, but clear recommendations have been limited by heterogeneity in measurement of brain health outcomes. We performed a scoping review to systematically evaluate brain health measurement in the scientific literature to date, informing development of a core outcome set.

**Design** Scoping review.

**Data sources** Medline, APA PsycArticles and Embase were searched through until 25 January 2023.

**Eligibility criteria for selecting studies** Studies were included if they described brain health evaluation methods in sufficient detail in human adults and were in English language.

**Data extraction and synthesis** Two reviewers independently screened titles, abstracts and full texts for inclusion and extracted data using Covidence software.

**Results** From 6987 articles identified by the search, 727 studies met inclusion criteria. Study publication increased by 22 times in the last decade. Cohort study was the most common study design (n=609, 84%). 479 unique methods of measuring brain health were identified, comprising imaging, cognitive, mental health, biological and clinical categories. Seven of the top 10 most frequently used brain health measurement methods were imaging based, including structural imaging of grey matter and hippocampal volumes and white matter hyperintensities. Cognitive tests such as the trail making test accounted for 286 (59.7%) of all brain health measurement methods.

**Conclusions** The scientific literature surrounding brain health has increased exponentially, yet measurement methods are highly heterogeneous across studies which may explain the lack of clinical translation. Future studies should aim to develop a selected group of measures that should be included in all brain health studies to aid interstudy comparison (core outcome set), and broaden from the current focus on neuroimaging outcomes to include a range of outcomes.

## INTRODUCTION

Brain health can be defined as the preservation of optimal brain integrity and mental and cognitive function at a given age in the absence of overt brain diseases that affect normal brain function.[1] The ageing

## STRENGTHS AND LIMITATIONS OF THIS STUDY

⇒ Broad search strategy developed after a preliminary search of the current evidence base.
⇒ Wide inclusion criteria to capture maximal number of relevant studies.
⇒ Protocol does not include description of risk of bias for included studies.
⇒ Non-English-language articles were excluded.

population in the world is increasing and the number of people aged over 60 years is expected to grow to 2 billion in 2050.[2]

The Global Burden of Disease Study 2013 demonstrated that neurological disorders are a leading cause of chronic disorders worldwide, and that the years lived with disability for all neurological disorders increased by 59.6% from 1990 to 2013 as people are living for longer. The years lived with disability for Alzheimer's disease alone increased by 91.8% from 1990 to 2013.[3] Ten years on, the burden of disease has increased even further. In May 2022, the WHO member states implemented a global action plan to improve healthcare and well-being of people living with neurological disorders and reduce mortality, morbidity and disability associated with these conditions.[4]

The time is ripe to invest in methods of improving and optimising brain health to maximise the population quality of life and minimise disability, disease and death related to neurological diseases.[1]

The research world has responded by launching many studies to trial interventions to preserve brain health, but the wide variation in the methods used to study brain health is limiting comparison between studies[5] and therefore recommendations for interventions that can potentially improve brain health.[6] This has led to wasteful research practices—including repetition of studies comparing similar interventions but measuring different outcomes.[5 7] There is no consensus on a set

**BMJ**

of brain health outcomes that would be meaningful and important to patients, nor is there one on how specific outcomes should be measured and reported. There is an urgent need to achieve a consensus in brain health reporting to encourage prevention, optimisation and potentially even treatment for neurological diseases.

We aimed to conduct a systematic scoping review to evaluate methods of brain health measurement in current literature. This would enable us to identify and group brain health measurement tools and evaluate patterns of use of specific tools based on study locations, study types and year of publication. Core outcome sets (COS) are agreed standardised sets of outcomes that should be measured and reported, as a minimum, in all clinical trials in specific areas of health or healthcare.[5] These could be extended to include other types of study design. Despite the introduction of organisations such as the Core Outcome Measures in Effectiveness Trials Initiative in 2010 and support from various organisations to boost COS use in research, COS uptake is low in many branches of research including brain health.[8] This scoping review can provide a useful overview of the current state of brain health research and provide a list of tools for brain health measurement that can be considered in COS development.

A scoping review was chosen as the best technique to perform an initial rapid mapping of current evidence on brain health and identify the most used brain health outcome measures, to inform future consensus work on brain health outcomes to facilitate development of a brain health COS.

## METHODS

This scoping review was conducted in accordance with the Preferred Reporting Items for Systematic Reviews and Meta-Analyses extension for scoping reviews (PRISMA-ScR) checklist.[9]

A preliminary search of Medline, the Cochrane Database of Systematic Reviews and JBI Evidence Synthesis was conducted, and no current or underway systematic reviews or scoping reviews were identified on this topic. Over 3000 papers were found on the preliminary Medline search with the search terms ("brain-health" OR "cognitive-health") AND ("measur*" OR "outcome*" OR "biomarker" OR "marker"), so there was sufficient evidence available to inform this review.

### Inclusion/exclusion criteria

Studies that met the following criteria were included in the review:
1. Participants must be human.
2. Participants must be aged 18 years or over.
3. Studies must report outcomes that are measuring 'brain health'.
4. Studies must be written in the English language.

Studies were excluded from the review if they did not report brain health measures with sufficient detail to

enable replication, for example, studies that reported that imaging was used without specifying fractional anisotropy as the measurement.

The human brain develops significantly between childhood and adulthood, with different structure, network organisation and function.[10 11] Studies about children or adolescents were excluded as brain health measurement tools in children may not be suitable for adults and vice versa. Brain health is a human concept due to the complexity of human brain functions; therefore, we excluded studies on non-humans.

### Search strategy

A systematic search was conducted of Medline, Embase and APA PsycArticles databases for articles published from the inception of each of these databases to 25 January 2023 using a search strategy developed with an information specialist (online supplemental appendix I). The syntax of the search strategy was modified for use with Embase and APA PsycArticles.

Due to the relatively new concept of brain health, the search strategy was informed by an initial limited search of Medline. The following search terms were used: ("brain-health" OR "cognitive-health") AND ("measur*" OR "outcome*" OR "biomarker" OR "marker") on 12 December 2022, and 2362 results were screened by one author, of which 72 full-text papers were found to be suitable for inclusion for the review. The Yale Medical Subject Headings (MeSH) analyser was used to extract all MeSH and author keywords used in these 72 full-text papers. The terms were analysed with RStudio (V.2022.12.0+353 (2022.12.0+353)). 1035 search terms were used in the 72 papers, with 286 distinct search terms. All terms were considered for inclusion into the search strategy (online supplemental appendix I).

This scoping review included all study designs, including experimental and quasi-experimental study designs such as randomised controlled trials, non-randomised controlled trials, before and after studies and interrupted time-series studies; analytical observational studies such as prospective and retrospective cohort studies, case–control studies and analytical cross-sectional studies; descriptive observational study designs such as case series, individual case reports and descriptive cross-sectional studies; systematic reviews, text and opinion papers and conference abstracts if they met the inclusion criteria.

### Source of evidence selection

Following the search on 25 January 2023, all identified citations were uploaded into Covidence (Covidence systematic review software, Veritas Health Innovation, Melbourne, Australia; available at https://www.covidence.org/.), which is a web-based collaboration software platform that streamlines the production of systematic and other literature reviews. Duplicates were removed by Covidence during this process, and further duplicates were manually removed.

Following a pilot test, AL and another independent reviewer (SS, KA, PI, NM, GN or VP) screened each title and abstract for assessment against the inclusion criteria for the review. Full-text articles for potentially relevant sources were imported into Covidence, and these were assessed in detail against the inclusion criteria by AL and another independent reviewer (SS, KA, PI, NM, GN or VP). Reasons for exclusion of sources of evidence at full text that did not meet the inclusion criteria were recorded by the system. Conflicts in reviewer opinion were all resolved through discussion, although an additional independent reviewer (AT) was available for adjudication.

The results of the search and the study inclusion process are presented in a PRISMA-ScR flow diagram[9] (online supplemental appendix II).

### Data extraction
Data were extracted from included papers by two independent reviewers using a data extraction template developed by the reviewers on Covidence (online supplemental appendix III). All conflicts were resolved through discussion before the data extraction process was finalised.

All brain health measurement methods were grouped into categories and tabulated based on frequency of use. Study location was determined from the methods section of each study, and if this was not mentioned or an international cohort was used, the country of the first author's institution was entered as the study location. Study location was not entered for narrative or systematic reviews.

### Patient and public involvement
None.

### RESULTS
A total of 6155 studies were included in the title and abstract screening after removing duplicates from the original search results. After abstract review, 924 studies were assessed for eligibility using the inclusion and exclusion criteria, leaving 727 studies for data extraction (online supplemental appendix II).

### Study types
There were 609 (83.8%) cohort studies, 59 (8.1%) randomised controlled trials or substudies within randomised controlled trials; 25 (3.4%) case series; 19 (2.6%) systematic reviews with or without meta-analyses; 11 (1.5%) narrative reviews and 4 (0.6%) were other study types.

There was a wide heterogeneity in brain health measurement methods between study types, and more than 60% of studies in each type used more than one modality (imaging, cognitive, mental health, clinical or biological) to measure brain health (online supplemental table 1). Mental health measurement methods were the least used category, used in only 11 cohort studies and 2 narrative reviews.

### Temporal trends
The range of years of publication of brain health studies was between 2003 and 2023. Online supplemental figure A shows a histogram of the number of brain health publications per year. The number of published brain health studies is steadily increasing and has more than tripled in the last 5 years (54 papers published in 2017 and 181 papers published in 2022), and increased by 22 times in the last 10 years (8 papers published in 2012).

The percentage of studies using mental health, clinical and biological methods to measure brain health has increased in the last 5 years, and the number of studies using multiple categories of brain health measurement has increased over time (online supplemental table 2).

### Geographical trends
Online supplemental figure B shows a heat map of the study location of the 697 brain health publications from this review (these data exclude systematic and narrative reviews). Online supplemental table 3 shows a list of the 39 countries where brain health studies were carried out and the number of studies using each category of brain health measurement. The USA alone accounts for almost 50% of all published brain health studies, with most studies published in the states of California, Massachusetts and Maryland (22%, 10% and 8% of all US studies, respectively). The top five countries researching brain health (in order) were the USA, the UK, Canada, Australia and China. Only 43 studies (6% of 697) took place in 11 low-income or middle-income countries (LMICs) defined by the World Bank, and the two studies that took place on the African continent were led by US or UK researchers. Half the studies in LMICs were multicategory studies, and the other half used imaging techniques as the sole method of brain health measurement. None of the LMIC studies used mental health or biological techniques to measure brain health.

### Brain health measurement methods
There were 478 unique methods of brain health measurement identified in the data extraction. Two hundred and sixty-eight (56.1%) of these were only used once. The remaining 210 methods will be presented in imaging, biological, clinical, mental health and cognitive test categories.

Within these categories, one study (0.1%) included outcome measures from four categories (cognitive, mental health, clinical and biological); 34 studies (4.7%) included measures from three categories (most commonly imaging, cognitive and biological); 233 studies (32.0%) included measures from two categories (most commonly imaging and cognitive); and the remaining 460 studies (63.3%) included measures from one of these categories (most commonly imaging).

Eight of the top 10 most prevalent methods for measuring brain health were imaging based (table 1). These were mainly volume estimates for grey and white matter in specific regions, particularly the hippocampus,

**Table 1** Top 10 most used measures for brain health measurements

| Measurement method | Category | Number of studies using this method (%) |
|---|---|---|
| Grey matter volume in specific region(s) | Imaging | 133 (18.3) |
| White matter hyperintensities | Imaging | 133 (18.3) |
| Total brain volume | Imaging | 132 (18.2) |
| Whole brain grey matter volume | Imaging | 106 (14.6) |
| Hippocampal volume | Imaging | 105 (14.4) |
| Fractional anisotropy | Imaging | 102 (14.0) |
| White matter volume in specific region(s) | Imaging | 95 (13.1) |
| Trail making test A and/or B | Cognitive testing | 86 (11.8) |
| Whole brain white matter volume | Imaging | 77 (10.6) |
| Mini-Mental Status Examination | Cognitive testing | 73 (10.0) |

Row data are not mutually exclusive as many studies used more than one category of methods.

and the whole brain; presence of white matter hyperintensities and fractional anisotropy. The trail making test (TMT) and Mini-Mental Status Examination (MMSE) were the other two most prevalent methods (table 1).

## Imaging

Imaging was the most common method of brain health measurement (514 studies, 70.7%), particularly MRI-based measures. Within imaging, measurements were divided into structural, functional, diffusion MRI parameters, compound imaging indices and miscellaneous forms of imaging (online supplemental table 4).

Approximately one-fifth of all studies in our review used structural MRI-based volumetric estimates, particularly of grey matter and hippocampal volumes, or looked for the presence of white matter hyperintensities. Seven per cent of studies looked at cerebral blood flow in specific regions of the brain using functional MRI techniques at rest or while performing tasks. Brain age gap calculations comparing an imaging estimate of brain age derived from various MRI parameters with a person's chronological age were used in 1.8% of studies. Positron emission tomography-measured amyloid load or presence was the most used (5.2%) type of non-MRI method to measure brain health.

## Cognitive tests

Three-hundred and thirty (45.4%) studies used a form of cognitive test when measuring brain health. The highest number of individual brain health measurement methods used more than once was in this category (115 of 210, 54.8%). Only named test batteries or tests described in sufficient detail for replication were included in the data extraction.

The Trail Making Test A (TMT) or B, Mini Mental State Examination (MMSE) and Stroop tests were the most used of all cognitive tests, with approximately one-tenth of all studies using one or more of these in evaluating brain health (online supplemental table 5).

## Biological

A hundred studies (13.8%) used biological sampling from serum or whole blood, cerebrospinal fluid or post-mortem brain tissue to measure brain health (online supplemental table 6). *ApoE4* genotyping was the most common brain health measurement method in this category, used in 5.6% of all studies in our review. Other commonly measured biomarkers included brain-derived neurotrophic factor (BDNF), neurofilament light from cerebrospinal fluid and tau protein levels.

## Clinical

Electroencephalography (EEG) was the most used clinical method of evaluating brain health (3.3% of studies) (online supplemental table 7). Several studies employed an EEG-derived brain age estimation software to measure brain health. The lifestyle score for brain health, which was a composite score comprising of 12 modifiable risk factors for dementia, was used in 1.7% of studies. Clinical diagnosis of dementia and hand grip strength were also used as indicators of brain health. Sleep quality indexes and health-related quality of life surveys were also included in this category.

## Mental health

Thirty-seven (5.1%) studies measured mental health outcomes as an indicator of brain health (online supplemental table 8). The most commonly used measure was the Baratt Impulsiveness Scale, followed by a number of screening tools for depressive and anxiety symptoms. Other, more rarely used measures included those designed to identify perceived stress, rumination and post-traumatic stress disorder symptoms.

## DISCUSSION

Brain health is an emerging research area. Studies about brain health have increased significantly in the last decade, predominantly in the form of cohort studies investigating brain health preservation. Evaluating brain health is

complex and there is currently no single test that can be used to fully characterise an individual's brain health. Our scoping review found that brain health is most evaluated via imaging modalities (70.7% of studies) and cognitive testing (45.4% of studies), and approximately one-third of all studies used a combination of these two categories of outcomes. Mental health, biological markers and clinical methods of brain health measurement are also used and can provide a more holistic view of an individual's brain health.

## Imaging

There are many reasons to use imaging parameters to measure brain health. MRI enables detailed, non-radioactive and non-invasive study of the structure, function and integrity of the brain with minimal risk to participants. Imaging studies can be performed using the same protocol on a large number of participants for cross-sectional comparison and repeated for longitudinal studies.[12] It is likely that most studies in our review chose brain volumetric measures to measure brain health due to the ease of obtaining volumetric data from structural MRI scans; the objective, easily interpreted and comparable nature of the data; and existing evidence that brain volume correlates with cognitive performance.[13 14] Structural information only represents the tip of the iceberg of information obtainable through an MRI scan—many scanners have the capability to also perform functional imaging at rest or during tasks; and diffusion imaging to explore brain microstructure[15] or other advanced applications such as spectroscopy.[16] Each of these modalities provides further information on different aspects of brain health, building a picture of overall brain structure, function and integrity. Increasing efforts toward collaboration[17 18] and building large biobanks of brain imaging datasets[19] have led to further innovation with MRI data, where models have been trained to predict brain age using population comparisons[20] or detect and score pathological changes to estimate brain health.

Another benefit of using imaging measurements for brain health includes the possibility of focusing on specific regions of the brain, or the presence or absence of specific abnormalities. Dementia is a condition that could be considered the antithesis to brain health, and many studies in our review studied the presence or absence of imaging markers of dementia to estimate brain health. Our findings that the most studied brain regions were the hippocampus and grey matter structures are consistent with the existing literature that these are the regions affected early in Alzheimer's disease.[21] Systematic review evidence shows that the presence and increasing volume of white matter hyperintensities on MRI are strongly associated with cognitive impairment and all-cause dementia.[22]

While MRI parameters have been a useful complement in dementia diagnosis, it is widely known that MRI appearances can be heterogeneous even for the same type of dementia and symptom burden.[23] Some imaging parameters that appear to be objective, such as fractional anisotropy, are still subject to some degree of human interpretation (to define boundaries, for example) and technical limitations such as the difficulty distinguishing crossing fibres from more structurally robust fibres.[24]

Another drawback of using MRI as a key method of measuring brain health is its significant cost. The UK's National Health Service estimates a cost per unit of MRI scan of one location without contrast as £146.75[25] (€170.51; $183.87), and another study estimated a diffusion-weighted MRI scan of the brain for patients with cholesteatoma to be in the region of $C390.66 (€266.63, $287.54).[26] These costs do not consider the cost of setup or maintenance of an MRI machine, or the potential need for specialist staff to run specific imaging protocols or interpret images from different modalities. The small number of brain health studies from LMICs may reflect difficulties in funding brain health research.

## Cognitive testing

Neuropsychological and cognitive tests assess a wide range of brain functions, including learning, reading, language and problem-solving skills,[27] providing useful insight into a person's cognitive ability. Many tests are cheap or freely available, easy to administer and can be performed in a clinic or home setting without requiring sophisticated equipment or a prolonged time. Some tests have been adapted into briefer versions or online versions,[28] or translated to a different language, further improving their reliability as 'universal' tools.[29] The TMT,[30] Stroop test,[31] Rey auditory verbal learning test,[32] MMSE[33] and Montreal Cognitive Assessment[34] were the top five most used cognitive assessments in our review. These are well-known, validated tests mainly used as clinical screening tools for cognitive impairment. Several cognitive tests such as the TMT have been adapted to be used during task-based functional MRI scans,[30] increasing the ease of combining imaging and cognitive testing when evaluating brain health.

Some limitations of using cognitive testing as the sole method to evaluate brain health include cost, limited sensitivity, ceiling effects and if used repeatedly can lead to bias due to learning. Tests such as the MMSE incur a copyright cost of approximately £0.80 (€1.00; $1.30) (2012 data),[35] not including the cost of supervision and test interpretation. Many tests rely on a baseline level of educational qualification or language, meaning that results are unreliable in those with a lower educational attainment or those with a different language or cultural background.[29] While technological advancements have enabled online or tablet assessments of cognition, problems such as computer anxiety and technological difficulties may limit their generalisability and reproducibility.[28] There has yet to be a consensus on a single cognitive test that provides a holistic view of cognitive function, and many studies evaluating and validating cognitive tests have methodological flaws such as small sample sizes, non-generalisable samples or conflicts of interests.

## Biological markers

Biological markers for brain health tend to be objective, quantifiable and repeatable measures. They can be an efficient method of measuring brain health as it is possible to gain information on various biomarkers using small volumes of blood or cerebrospinal fluid, and measures can be more easily compared across laboratories if the same protocols are used for sample processing and measurement. *ApoE4,* the most common genetic risk factor for Alzheimer's disease, was the most studied biological marker in our review, likely relating to its use as a predictor for poorer brain health but also its potential as a therapeutic target.[36] BDNF, the second most commonly studied biological marker, has been studied as a protective factor and therefore therapeutic target for a wide range of neurological conditions, including those relating to neurodegeneration and mental illness.[37] The disadvantages of using biological markers include their invasive nature, problems with sensitivity and specificity and associated cost of performing procedures, advanced laboratory methods, equipment and interpretation.

## Mental health symptom screening

The main advantages of incorporating mental health screening when evaluating brain health include the opportunity for early detection and treatment of common mental health conditions such as depression[38] and anxiety[39]; distinguish symptoms due to poor brain health from those relating to poor mental health[40]; and promote evidence-based practices that encourage better mental health which can in turn contribute to improving brain health, such as exercise[41] and sleep interventions.[42] Challenges researchers may encounter when implementing mental health screening include their self-reported nature, false positives, resource constraints and difficulty performing exhaustive screening for all conditions or selecting specific tests.[39] Mental health screening tools suffer from similar methodological issues as cognitive tests—validation is inconsistent across populations, cultures, educational background; some are subject to assessor or performance bias; and there is no single gold-standard test available that can measure a person's mental health.[43]

## Other methods

Other methods of brain health evaluation such as clinical diagnoses, EEG and lifestyle or patient-reported brain health scores are all potentially useful methods to measure other aspects of brain health but are all subject to bias and problems with reproducibility, cost and practical issues.

## Strengths and limitations

This is the first review to collate methods of brain health measurement in current literature, providing evidence of rapidly increasing interest in the field over the last decade and identifying the most used brain health outcome measures. This study clearly demonstrates the wide variation in outcome measures and lack of patient-reported outcomes used in brain health research and emphasises the need for outcome set development in this field.

The scoping nature of the review precluded detailed analyses of reasons behind outcome choices and risk of bias in each study. Bias may have been introduced by the lack of standardisation of brain health terminology and definitions during the search and screen, exclusion of non-English-language papers and the use of only three databases. The use of several independent reviewers and software reduced the risk of bias during the screening process. The databases and search terms were chosen after extensive consideration and discussion with an independent data specialist within the University of Oxford, and the protocol was reviewed by an experienced investigator with extensive experience with Cochrane reviews.

Measurement techniques such as MRI-derived volumetric estimations have evolved over the past decade, and new techniques such as machine learning to perform brain age calculations have only recently been developed, limiting the utility of considering frequency of use as the main outcome measure in our review.

## CONCLUSIONS AND FUTURE DIRECTIONS

Brain health has become an increasingly popular topic of research and is most frequently evaluated using imaging parameters, alongside other measures such as cognitive testing, biological markers, mental health testing and clinical tests.

Future work should focus on fine-tuning brain health definitions and engaging stakeholders and experts to develop a COS for brain health studies that can be informed by findings from this review. There is an urgent need for a COS in this field to facilitate cross-study comparisons, particularly for interventional studies to improve or maintain brain health. Outcomes should broaden the focus from expensive neuroimaging methods to encompass a more holistic view of the brain, for example, mental health outcomes that are currently neglected in the literature. Consensus work involving patients, carers and professionals should be undertaken to ensure the core outcomes are useful and relevant.

**Contributors** AL designed the protocol, led the study, performed the search, first reviewed all abstracts and full texts and extracted data for the work, drafted the manuscript, made edits and submitted the manuscript. SS, KA, PI, NM, GN and VP were second reviewers for all abstracts and full texts and extracted data for the work, reviewed the manuscript and provided final approval for the publication. AT, RSM and NR provided substantial contributions to the conception and design of the study, reviewed the protocol and manuscript, and provided final approval for the publication. All authors agree to be accountable for all aspects of the work in ensuring that questions related to the accuracy or integrity of any part of the work are appropriately investigated and resolved. Dr Kapil Savjani provided help with second reviewing for the abstracts and full texts. AL is responsible for the overall content of this article as a guarantor.

**Funding** AL is supported by a clinical doctoral research scholarship from the Nuffield Department of Population Health, University of Oxford. AT is supported by a Wellcome Trust fellowship (216462/Z/19/Z).

**Competing interests** None declared.

**Patient and public involvement** Patients and/or the public were not involved in the design, or conduct, or reporting, or dissemination plans of this research.

**Patient consent for publication** Not required.

**Ethics approval** Not applicable.

**Provenance and peer review** Not commissioned; externally peer reviewed.

**Data availability statement** Data sharing not applicable as no datasets generated and/or analysed for this study.

**ORCID iDs**
Angeline Lee http://orcid.org/0000-0001-6068-2689
Reem Saleem Malouf http://orcid.org/0000-0002-0673-5126
Anya Topiwala http://orcid.org/0000-0002-8408-0372

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
