## [Reviewer comments · BMJ Open]

ARTICLE DETAILS

TITLE (PROVISIONAL)	Brain health measurement – a scoping review
AUTHORS	Lee, Angeline; Shah, Suraj; Atha, Karyna; Indoe, Peter; Mahmoud, Naira; Niblett, Guy; Pradhan, Vidushi; Roberts, Nia; Malouf, Reem; Topiwala, Anya

VERSION 1 – REVIEW

REVIEWER	Rubin, Michael The University of Texas Southwestern Medical Center, Neurology
REVIEW RETURNED	11-Oct-2023

GENERAL COMMENTS	The authors succeed in their goal to review an expansive topic and provide a useful summary of the science to date. A detailed copy editing process would find a handful of minor errors. I would like to see more of an exploration of the need to review the topic and suggestions for how to move the field forward in the discussion/conclusion.
--

REVIEWER	Rosendale, Nicole University of California San Francisco
REVIEW RETURNED	25-Oct-2023

GENERAL COMMENTS	I commend the authors for a rigorous methodology in evaluating the current landscape of brain health research, and for screening so many articles - I recognize that that is a lot of work! I had a few questions / edits listed below:  1) In the methods section of the abstract, recommend including the date ranges for the included articles (e.g., studies published between X date and X date were included). 2) Consider changing the wording in the abstract conclusion section regarding core out set development. For those not familiar with that term, the meaning is not clear as the sentence is currently written. 3) In the "Search strategy" section, recommend defining the date for "present" (line 32). 4) Use of the term "considered" in the 3rd paragraph of the "Search strategy" section is confusing. Does this mean that all the listed study types were included if they met the inclusion criteria listed above? If so, this paragraph can state this much more directly. Considered makes it sound as if there were additional, not-specified criteria used to determine inclusion (and if that is the case, this should be detailed). 5) Please include the initials of the two reviewers in the text. 6) In the tables in the results section, it would be helpful to signify with a footnote on the tables that the rows are not mutually exclusive (so the % will add up to >100). Is the % associated with
---

	the number of studies using the method out of the entire number of articles or the number of articles that used that particular method? 7) I would not characterize the PSQI as a measure of mental health but rather sleep. Similarly, I question if HRQoL is a mental health outcome as it measures a number of factors. 8) I'm curious if there were particular methods used in the interventional studies vs. the observational? Is there more consensus on methods amongst the RCT data, for example? Additionally, are there particular geographic patterns? The method chosen may be limited by availability of particular services (MRI for example). Similarly, are there temporal trends? Are we coming closer to consensus if looking at the methods in the past 5 years, for example? 9) Recommend discussing cost considerations as a potential limitation for MRI based outcomes (similar to what is mentioned for cognitive testing). 10) I'm interested in the inclusion of studies that had mental health as the primary brain health method as we commonly think of mental health as a potential variable within the pathway between a particular exposure and brain health outcome. I would think these would not qualify for inclusion in the scoping review based on your focus on brain health (and not, for example, mental health). If word count allows, I recommend describing these papers in a bit more detail to help readers understand how they connected mental health outcomes to brain health.
--	---

VERSION 1 – AUTHOR RESPONSE

Reviewer: 1

Dr. Michael Rubin, The University of Texas Southwestern Medical Center

Comments to the Author:

The authors succeed in their goal to review an expansive topic and provide a useful summary of the science to date. A detailed copy editing process would find a handful of minor errors. I would like to see more of an exploration of the need to review the topic and suggestions for how to move the field forward in the discussion/conclusion.

Thank you, we have reviewed the manuscript for errors and added more detail in the introduction and discussion sections. Specifically, we have added more detail about the utility of the work in the introduction and suggestions for how to move the field forward in the discussion. The changes we have made are in the following two paragraphs.

This would enable us to identify and group brain health measurement tools and evaluate patterns of use of specific tools based on study locations, study types and year of publication. Core outcome sets (COS) are agreed standardised sets of outcomes that should be measured and reported, as a minimum, in all clinical trials in specific areas of health or healthcare(5). These could be extended to include other types of study design. Despite the introduction of organisations such as the Core Outcome Measures in Effectiveness Trials (COMET) initiative in 2010 and support from various organisations to boost COS use in research, COS uptake is low in many branches of research including brain health(8). This scoping review can provide a useful overview of the current state of brain health research and provide a list of tools for brain health measurement that can be considered in COS development.

'There is an urgent need for a COS in this field to facilitate cross-study comparisons, particularly for interventional studies to improve or maintain brain health. Outcomes should broaden the focus from expensive neuroimaging methods to encompass a more holistic view of the brain, for example mental health outcomes that are currently neglected in the literature. Consensus work involving patients, carers and professionals should be undertaken to ensure the core outcomes are useful and relevant.'

Reviewer: 2

Dr. Nicole Rosendale, University of California San Francisco

Comments to the Author:

I commend the authors for a rigorous methodology in evaluating the current landscape of brain health research, and for screening so many articles - I recognize that that is a lot of work!

Many thanks!

I had a few questions / edits listed below:

1) In the methods section of the abstract, recommend including the date ranges for the included articles (e.g., studies published between X date and X date were included).

We have now added this on page 1 as follows: Medline, APA PsycArticles and Embase were searched through till 25th January 2023.

2) Consider changing the wording in the abstract conclusion section regarding core out set development. For those not familiar with that term, the meaning is not clear as the sentence is currently written.

Thank you for this suggestion. We now include the following explanation on page 1: Future studies should aim to develop a selected group of measures that should be included in all brain health studies to aid inter-study comparison (core outcome set); and broaden from the current focus on neuroimaging outcomes to include a range of outcomes.

3) In the "Search strategy" section, recommend defining the date for "present" (line 32).

We have now added this on page 4 as follows:

"A systematic search was conducted of Medline, Embase and APA PsycArticles databases for articles published from the inception of each of these databases to 25th January 2023."

4) Use of the term "considered" in the 3rd paragraph of the "Search strategy" section is confusing. Does this mean that all the listed study types were included if they met the inclusion criteria listed above? If so, this paragraph can state this much more directly. Considered makes it sound as if there were additional, not-specified criteria used to determine inclusion (and if that is the case, this should be detailed).

Apologies for the confusion. The terminology has been changed to explicitly state that these study types were included if they met the inclusion criteria. Page 4-5:

“This scoping review included all study designs, including experimental and quasi-experimental study designs such as randomized controlled trials, non-randomized controlled trials, before and after studies and interrupted time-series studies; analytical observational studies such as prospective and retrospective cohort studies, case-control studies and analytical cross-sectional studies; descriptive observational study designs such as case series, individual case reports and descriptive cross-sectional studies; systematic reviews, text and opinion papers and conference abstracts if they met the inclusion criteria.”

5) Please include the initials of the two reviewers in the text.

“Two independent reviewers” changed to “AL and another independent reviewer (SS, KA, PI, NM, GN, or VP)” for clarity. Page 5.

6) In the tables in the results section, it would be helpful to signify with a footnote on the tables that the rows are not mutually exclusive (so the % will add up to >100). Is the % associated with the number of studies using the method out of the entire number of articles or the number of articles that used that particular method?

Added “**Rows in the table are not mutually exclusive as many studies used more than one method of measurement.*” As a footnote for every table.

Added (% of total studies) in column heading for every table.

All the tables except one showing the top ten most common brain health research methods have been moved to the supplementary information document due to formatting restrictions.

7) I would not characterize the PSQI as a measure of mental health but rather sleep. Similarly, I question if HRQoL is a mental health outcome as it measures a number of factors.

Thank you for highlighting this. The PSQI and HRQoL have now been moved into a ‘Clinical’ category as these are more lifestyle-based measures. The accompanying text in results section has been amended accordingly.

8) I'm curious if there were particular methods used in the interventional studies vs. the observational? Is there more consensus on methods amongst the RCT data, for example? Additionally, are there particular geographic patterns? The method chosen may be limited by availability of particular services (MRI for example). Similarly, are there temporal trends? Are we coming closer to consensus if looking at the methods in the past 5 years, for example?

These were interesting points and we've explored all of them and added new sections in results to discuss on page 6: interventional vs observational studies, geography, and temporal trends. We have also added supplemental tables and figures to demonstrate this data.

9) Recommend discussing cost considerations as a potential limitation for MRI based outcomes (similar to what is mentioned for cognitive testing).

Thank you, we now do this on page 14, as follows:

“Another drawback of using MRI as a key method of measuring brain health is its significant cost. The United Kingdom’s national health service (NHS) estimates a cost per unit of MRI scan of one location without contrast as £146.75(25) (€170.51; \$183.87), and another study estimated a diffusion weighted MRI scan of the brain for cholesteatoma patients to be in the region of \$390.66 Canadian dollars (€266.63, \$287.54)(26). These costs do not consider the cost of setup or maintenance of an MRI machine, or the potential need for specialist staff to run specific imaging protocols or interpret images from different modalities. The small number of brain health studies from low to middle income countries may reflect difficulties in funding brain health research.”

10) I'm interested in the inclusion of studies that had mental health as the primary brain health method as we commonly think of mental health as a potential variable within the pathway between a particular exposure and brain health outcome. I would think these would not qualify for inclusion in the scoping review based on your focus on brain health (and not, for example, mental health). If word count allows, I recommend describing these papers in a bit more detail to help readers understand how they connected mental health outcomes to brain health.

There were no studies that used mental health parameters as the sole or primary brain health outcome. MH parameters were always included as an adjunct to another type of study, e.g. imaging.[MOU1] The broad definition of brain health (“the preservation of optimal brain integrity and mental and cognitive function at a given age in the absence of overt brain diseases that affect normal brain function”) has led to studies including mental health outcomes under this umbrella.

[MOU1]This is the only point which you haven't totally addressed it. Could you say something like brain health is broadly defined and some include mental health outcomes as under that umbrella? Is there some definition of brain health somewhere that includes mental health that you could help justify?

VERSION 2 – REVIEW

REVIEWER	Rosendale, Nicole University of California San Francisco
REVIEW RETURNED	23-Jan-2024
GENERAL COMMENTS	The authors have appropriately addressed the reviewers' concerns.